# Sperm-inherited H3K27me3 impacts offspring transcription and development in *C. elegans*

Kiyomi Raye Kaneshiro[1], Andreas Rechtsteiner[1] & Susan Strome [1]

Paternal epigenetic inheritance is gaining attention for its growing medical relevance. However, the form in which paternal epigenetic information is transmitted to offspring and how it influences offspring development remain poorly understood. Here we show that in *C. elegans*, sperm-inherited chromatin states transmitted to the primordial germ cells in offspring influence germline transcription and development. We show that sperm chromosomes inherited lacking the repressive histone modification H3K27me3 are maintained in that state by H3K36me3 antagonism. Inheritance of H3K27me3-lacking sperm chromosomes results in derepression in the germline of somatic genes, especially neuronal genes, predominantly from sperm-inherited alleles. This results in germ cells primed for losing their germ cell identity and adopting a neuronal fate. These data demonstrate that histone modifications are one mechanism through which epigenetic information from a father can shape offspring gene expression and development.

---

[1] Department of Molecular, Cell and Developmental Biology, University of California, Santa Cruz, 1156 High Street, Santa Cruz, CA 95064, USA. Correspondence and requests for materials should be addressed to S.S. (email: sstrome@ucsc.edu)

There is growing awareness that development and health are influenced by epigenetic information passed from parents to offspring[1,2]. The gametes, sperm and oocyte, are the conduits through which epigenetic information is passed from one generation to the next. Investigating epigenetic inheritance via the maternal line is complicated by the many non-epigenetic factors the oocyte provides to the embryo[1]. For this reason, the field of epigenetic inheritance largely focuses on transmission via sperm. DNA methylation and non-coding RNAs (ncRNAs) are epigenetic carriers that are transmitted by sperm to the embryo and that have been shown to influence gene expression in offspring[3,4]. Modified histones can also be passed, but their role in intergenerational and transgenerational epigenetic inheritance is less clear. In mammals, the sperm genome is primarily packaged with protamines and, while there is current debate about the degree to which the sperm genome retains histones, there is consensus that marked histones are found on developmentally important loci[5–7]. Despite this observation and mounting suggestive evidence[8–10], establishing a causal role for sperm-inherited histone marks in regulating offspring transcription and development has been an ongoing challenge. We took advantage of features of *C. elegans* to tackle this issue.

We previously showed that *C. elegans* sperm retain nucleosomes and histone marking genome-wide[11] and that *C. elegans* Polycomb Repressive Complex 2 (PRC2) maintains inherited states of H3K27me3 during embryogenesis[12]. In wild-type embryos H3K27me3 is enriched over genes that were silent in the parental germline[13,14]. H2K27me3 marking inherited from hermaphrodite parent worms is essential for germline development in offspring, since hermaphrodite parents lacking PRC2 generate offspring in which the primordial germ cells die during early larval development[15]. We thus hypothesize that transmission of chromatin states from parent germ cells via sperm and oocyte to the nascent germ cells in offspring protects germline-appropriate gene expression patterns in the developing and adult germline.

In this work we investigate how chromatin states inherited from parents are maintained in offspring and whether inherited states are important for offspring transcription and development. We elucidate a mechanism through which an inherited H3K27me3(−) state is propagated from parent germ cells (sperm) to offspring germ cells. We show that inheriting a sperm genome lacking the repressive mark H3K27me3 results in derepression of many genes for somatic development, especially neuronal genes, in offspring germlines. This results in germ cells that in a sensitized genetic background lose their germ cell identity and adopt a neuronal fate. Taken together, these findings establish a cause–effect relationship between sperm-inherited histone marks and offspring transcription and development in *C. elegans*.

## Results

**Maintenance of an inherited H3K27me3(−) state requires MES-4.** When *C. elegans* embryos inherit some chromosomes with and some chromosomes without H3K27me3, PRC2 maintains inherited states by (1) restoring levels of H3K27me3 on H3K27me3(+) chromosomes after DNA replication and (2) failing to de novo methylate H3K27me3(−) chromosomes[12]. The ability of PRC2 to restore levels of H3K27me3 after genome duplication is likely explained by the EED subunit of PRC2 (MES-6 in worms) binding to H3K27me3 and stimulating the methyltransferase activity of the EZH2 subunit (MES-2 in worms)[16,17]. How PRC2 is prevented from de novo methylating chromosomes inherited lacking H3K27me3 is less clear. One possibility is that chromosomes lacking H3K27me3 are unable to

recruit and stimulate PRC2 activity. Another possibility is that these chromosomes bear an opposing mark that antagonizes PRC2 activity. Likely candidates for antagonizing PRC2 activity are histone marks associated with gene expression, hereafter referred to as active marks, and their corresponding histone modifiers[18,19].

To test if active marks prevent PRC2 from de novo methylating sperm-inherited H3K27me3(−) chromosomes during early embryogenesis, we monitored sperm chromosomes through rounds of cell division in embryos lacking a maternal load of MES-4 or SET-2, which generate the active marks H3K36me2/3[13,20] and H3K4me2/3[21], respectively. We generated embryos that inherited H3K27me3(+) oocyte chromosomes and H3K27me3(−) sperm chromosomes by mating wild-type females with PRC2 mutant males (Fig. 1a). To increase the likelihood that sperm chromosomes completely lacked H3K27me3, we used PRC2 mutant males whose parents also lacked PRC2. We call the offspring of wild-type mothers and PRC2 mutant fathers *K27me3 M+P−* (M+ for maternal H3K27me3(+), P- for paternal H3K27me3(−)). We assessed whether sperm-inherited chromosomes retained the H3K27me3(−) state or acquired H3K27me3, by tracking the status of H3K27me3 on sperm-inherited chromosomes during early embryogenesis. To facilitate analysis of sperm vs. oocyte chromosomes, we performed this analysis in a temperature-sensitive *plk-1* mutant background that keeps sperm-inherited and oocyte-inherited chromosomes in separate nuclei for many divisions[22] (Fig. 1b).

In all 2-cell *K27me3 M+P− plk-1* embryos, we observed that one nucleus in each cell contained all H3K27me3(+) chromosomes (oocyte-inherited) and the other nucleus contained H3K27me3(−) chromosomes (sperm-inherited). In control and SET-2-lacking embryos, this pattern persisted in all cells during the early divisions and in the P lineage (germ lineage) throughout embryogenesis (Fig. 1c, d, Supplementary Fig. 1). In contrast, in embryos depleted of MES-4, we observed gradual acquisition of H3K27me3 on the entire set of sperm chromosomes (Fig. 1c, Supplementary Fig. 1). To assess the appearance of H3K27me3 on sperm chromosomes, we analyzed *K27me3 M+P-* embryos from mothers that were either mutant or wild type for *mes-4* (see methods). At the 1-cell stage, all *K27me3 M+P−* embryos examined contained sperm chromosomes lacking detectable H3K27me3 irrespective of maternal MES-4 ($n = 6$ with maternal MES-4 and $n = 4$ without maternal MES-4). In the presence of maternal MES-4, this state was maintained in all embryos up to the 4-cell stage ($n = 21$) and in most embryos up to the 40-cell stage ($n = 15/19$). In the absence of maternal MES-4, all 2-cell embryos examined acquired some H3K27me3 on sperm chromosomes ($n = 6$). The levels of H3K27me3 on sperm and oocyte chromosomes became indistinguishable in embryos by the 4-cell stage ($n = 4$) and at later stages ($n = 15$). This gradual acquisition of H3K27me3 matches the gradual loss of H3K36me3 in the absence of maternal MES-4[23]. Thus, MES-4 inhibits PRC2 activity, presumably through MES-4-mediated methylation of H3K36, to maintain the paternal H3K27me3(−) state. These findings show that the H3K27me3(−) state is maintained by opposing PRC2 activity with active marks and also suggest that PRC2 does not require H3K27me3 to seed its activity, as observed in mouse embryonic stem cells[24].

**Sperm chromatin states shape transcription in offspring.** Do gamete-inherited chromatin states influence gene expression in offspring? This is a burning question in the field and one that remains elusive, in part because it is difficult to determine cause versus consequence when comparing chromatin states with transcriptional status. To establish a cause–effect relationship

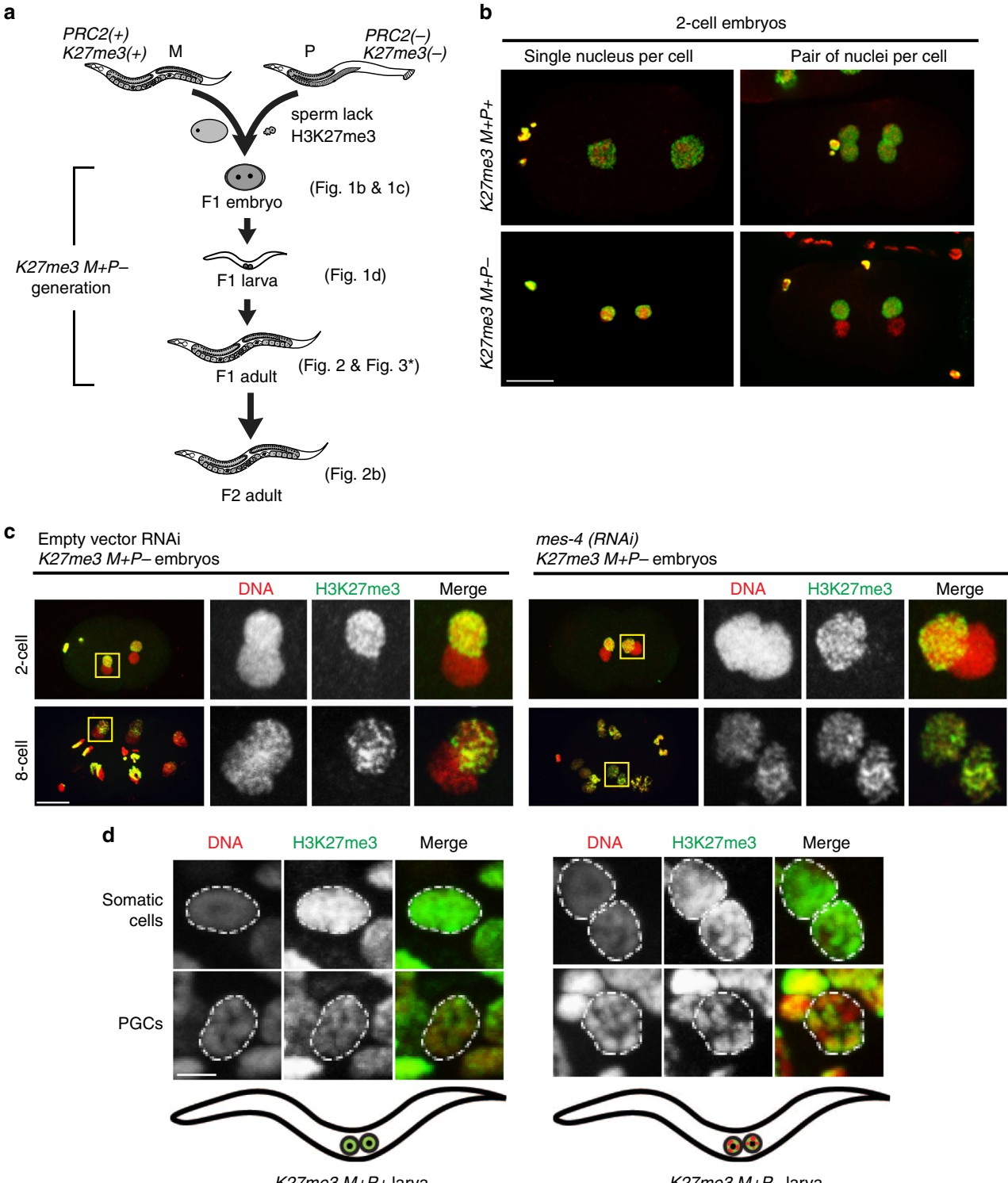

**Fig. 1** Maintenance of an inherited H3K27me3(−) chromatin state depends on antagonism of PRC2 by MES-4. **a** Diagram showing the cross between wild-type *PRC2(+) K27me3(+)* females and *mes-3/mes-3 PRC2(−) K27me3(−)* males that generates *K27me3 M+P−* F1 offspring. Maternal and paternal pronuclei are shown in the F1 1-cell embryo. The 2 primordial germ cells (PGCs) are shown in the newly hatched F1 larva. F1 hermaphrodites self-cross to produce F2 offspring. Figures that analyzed each generation/stage are indicated in parentheses. M, maternal; P, paternal; *Sensitized genetic background. Hybrid F1s were generated by mating Hawaii mothers with Bristol fathers. **b** Images of wild-type 2-cell embryos (left panels) or *plk-1* 2-cell embryos (right panels) demonstrate that the H3K27me3(−) state of sperm chromosomes in *K27me3 M+P−* embryos (bottom panels) is easily monitored when sperm- and oocyte-inherited chromosomes are kept in separate nuclei in *plk-1* mutants. DAPI-stained DNA in red. H3K27me3 immunostaining in green. Scale bar represents 10 μm. **c** 2-cell and 8-cell stage *K27me3 M+P−* embryos whose mothers were fed empty vector RNAi (control) or *mes-4(RNAi)* to knock down the maternal load of MES-4. DAPI-stained DNA in red. H3K27me3 immunostaining in green. Regions boxed in yellow in the left panels are shown at higher magnification in the right panels. Scale bar represents 10 μm. **d** Somatic cells and PGCs in a *K27me3 M+P+* L1 (left) and a *K27me3 M+P−* L1 (right). L1 schematics show the observed staining pattern in PGCs for each genotype. DAPI-stained DNA in red. H3K27me3 immunostaining in green. Scale bar represents 2 μm

between inherited chromatin and offspring transcription requires that (1) genetically identical individuals display different transcriptional outcomes as a result of inheriting different chromatin states, (2) transcriptional changes must be demonstrated to occur in *cis*, thus eliminating alternative epigenetic carriers that function in *trans* (e.g., cytoplasmic ncRNAs), and (3) DNA methylation, which also functions in *cis*, must be eliminated as a potential mediator of the observed transcriptional changes. We met the third criterion by using the model system *C. elegans*, which lacks canonical DNA methylation. We met the remaining criteria by comparing transcription from sperm-inherited and oocyte-inherited alleles in genetically identical (hybrid *fem-2/+*; *mes-3/+*) worms (Fig. 1a) that inherited the sperm genome with or without H3K27me3, which we call *K27me3 M+P+* and *K27me3 M+P−*.

We showed previously that *K27me3 M+P−* embryos have different chromatin states on their sperm-inherited and oocyte-inherited genomes and that those states are maintained in all cells of early embryos[12]. As embryos develop, the gamete-inherited chromatin states are rewritten in somatic cells but maintained through the five cell divisions that generate the two primordial germ cells (PGCs) present in newly hatched larvae[12] (Fig. 1d). When newly hatched larvae begin to feed, the PGCs launch their transcriptional program and begin to proliferate. During larval development, H3K27me3 gradually begins to accumulate on sperm-inherited chromosomes[12]. Absence of H3K27me3 on sperm chromosomes at the onset of germ cell transcription could result in the activation of inappropriate genes. Once activated, a euchromatic state could be propagated at specific loci, sensitizing daughter germ cells to continued expression of inappropriate genes and potentially downstream target genes. We investigated whether global differences in inherited chromatin states in PGCs impact transcription in the developing germline. We used single-nucleotide polymorphisms (SNPs) between parental strains (from Bristol, England, and from Hawaii) to compare transcription from sperm-inherited and oocyte-inherited genomes. This allowed us to determine whether transcriptional changes occurred in *cis* (from sperm alleles specifically) or in *trans* (from both sperm and oocyte alleles).

Worms that inherited sperm chromosomes lacking H3K27me3 misexpressed genes in their germlines; 149 genes were upregulated and 116 genes were downregulated (FDR < 0.1) in *K27me3 M+P−* compared to *K27me3 M+P+* worms (Fig. 2a, Supplementary Figs. 2, 3). Of the genes that were upregulated in *K27me3 M+P−* germlines, 96% are enriched for H3K27me3 in wild-type sperm[11] (Supplementary Fig. 4). While *K27me3 M+P−* worms were typically fertile, they generated a significant proportion of sterile offspring (Fig. 2b). These findings demonstrate that altering sperm-inherited chromatin states results in changes to germline transcription and function.

Expression of genes assessed by reads overlapping SNPs between Bristol and Hawaii correlated well with expression assessed by using all reads mapping to a gene (Supplementary Fig. 3a). Therefore, we could use SNPs to determine, for the 23% of misexpressed genes that contain a SNP, if transcription differences arose from the sperm allele, the oocyte allele, or both (Supplementary Data 1). Our analysis of SNP-containing transcript reads indicated that downregulated genes reflect decreased transcription from both the sperm and oocyte allele. However, upregulated genes reflect increased transcription primarily from the sperm allele (Fig. 2c, d). Importantly, approximately half of these genes were upregulated in *cis* with increased transcription from the sperm allele specifically (dark red triangles in Fig. 2c). These findings demonstrate that differential marking on sperm-inherited and oocyte-inherited genomes allows sperm-inherited and oocyte-inherited alleles

within the same cell to achieve different transcriptional outcomes in offspring. Given that *C. elegans* lacks canonical DNA methylation, this finding establishes a causal role for sperm-inherited H3K27me3 in regulating transcription in offspring. It also raises the possibility that environmentally induced changes to gamete marking could lead to transcriptional changes in offspring, with developmental consequences.

The genes upregulated from both sperm and oocyte alleles likely represent downstream targets of an initially misexpressed factor(s). The observation that these genes are more upregulated from sperm alleles than oocyte alleles led us to hypothesize that the presence of H3K27me3 on the oocyte genome acts as a barrier against transcriptional activation of germline-inappropriate genes. H3K27me3 on the oocyte-inherited genome would not be expected to protect it from factors that would shut down expression of germline-appropriate genes. This may explain the observation that downregulated genes were downregulated from both sperm and oocyte alleles. This led us to wonder if the global pattern of misexpression resulting from altered sperm chromatin was a concerted deviation from a germline program. To explore this, we performed gene ontology analysis on upregulated and downregulated genes. We found that downregulated genes are enriched for genes involved in reproduction and that upregulated genes are enriched for genes involved in somatic development, especially neuronal development (Supplementary Fig. 2b, c). This suggests that *K27me3 M+P−* germlines are deviating from a germline program of gene expression toward a somatic program, perhaps favoring a neuronal program.

**Sperm-inherited H3K27me3 protects germ cell fate in offspring.** To test whether germlines in *K27me3 M+P−* worms that inherited altered sperm chromatin are in fact transitioning toward a neuronal fate, we analyzed worms carrying a pan-neuronal reporter gene, *unc-119::GFP*. We compared expression of this reporter in genetically identical worms that inherited sperm chromatin with either a wild-type pattern of H3K27me3, *K27me3 M+P+*, or lacking H3K27me3, *K27me3 M+P−*. As noted above, *K27me3 M+P−* and *K27me3 M+P+* worms are fertile; we observed that fertile germlines do not detectably express *unc-119::GFP* (n > 500) nor do they significantly upregulate *unc-119* transcript. Performing this analysis in a sensitized genetic background in which the maternal load of PRC2 was reduced by half (*mes-3/+* heterozygous mother) rendered ~30% of *K27me3 M+P−* worms sterile (n = 267/894); <0.3% of *K27me3 M+P+* worms were rendered sterile (n = 2/870). In this background, we found that the *unc-119::GFP* neuronal reporter was expressed in a significant percentage of sterile *K27me3 M+P −* adult germlines (Supplementary Table 1). Notably, expression of the neuronal reporter increased with age, from 16% in sterile day 1 adults to 31% in sterile day 2 adults. Expression of the neuronal reporter was rarely detected (<0.1%) in germlines from late-stage larvae (L4) (n = 1/146) and never observed in the germlines from the rare sterile *K27me3 M+P+* worms. These findings indicate that deviation from a germline program in *K27me3 M+P−* worms is progressive and strongly suggest that loss of H3K27me3 from the sperm-inherited genome results in a failure to maintain germ cell identity rather than a failure to establish it.

To test whether neuronal genes are derepressed from endogenous loci in *K27me3 M+P−* germlines, we stained UNC-119::GFP(+) *K27me3 M+P−* germlines for UNC-64, a neuronally expressed plasma membrane protein involved in synaptic vesicle fusion[25]. We detected this protein in all of the germlines analyzed (n = 6) (Fig. 3c). To test for loss of expression of genes involved in reproduction, we stained sterile *K27me3*

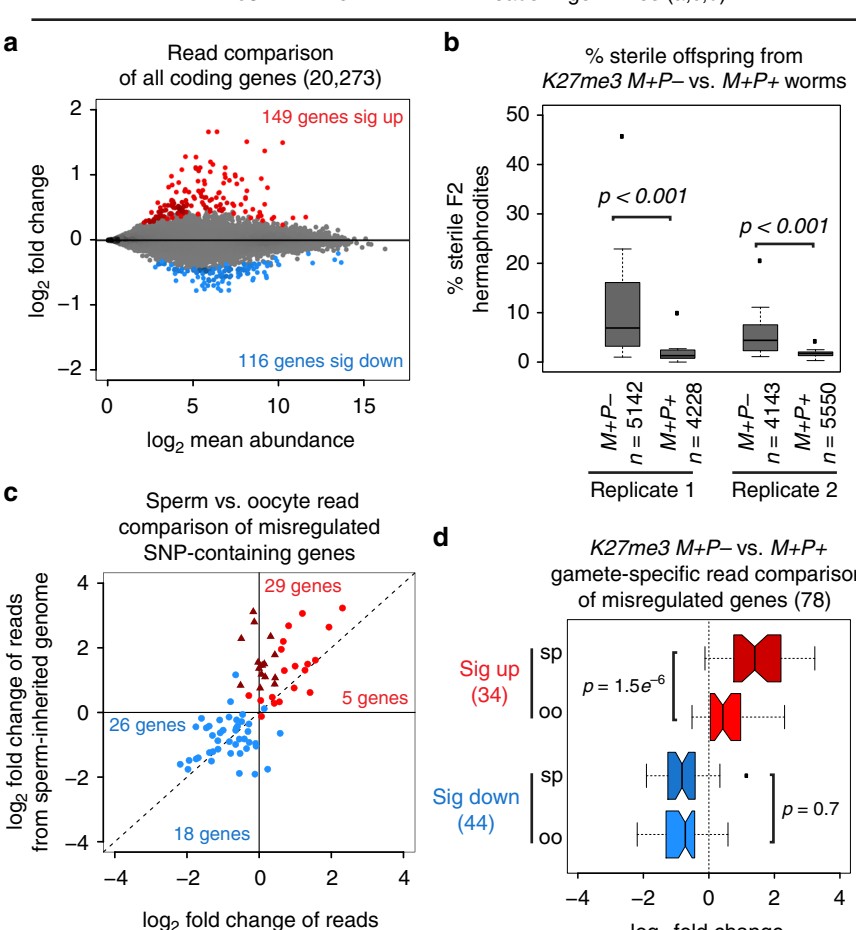

**Fig. 2** Inheritance of sperm chromosomes lacking H3K27me3 alters germline transcription and function in offspring. **a** MA plot comparing germline transcripts (RNA-seq reads) from *K27me3 M+P−* hermaphrodites to genetically identical *K27me3 M+P+* hermaphrodites. Genes that are significantly upregulated or downregulated (*p* < 0.1) are highlighted in red and blue, respectively. **b** Boxplots showing percent sterility of the F2-generation hermaphrodites from *K27me3 M+P−* and *K27me3 M+P+* F1 worms for each of 2 replicates. *n* = the number of F2 worms scored. Each box extends from the 25th to the 75th percentile, with the median indicated by the horizontal line; whiskers extend to the most extreme data point which is no more than 1.5 times the interquartile range from the box; individual data points outside of these boundaries are shown as filed boxes. *p*-values were calculated using a Wilcoxon rank sum test. **c** Scatter plot comparing sperm-specific and oocyte-specific log$_2$ fold changes of SNP-containing reads from significantly upregulated genes (in red) and downregulated genes (in blue). Genes upregulated in *cis* (>1.5-fold increase in reads from the sperm allele; <1.5-fold increase from the oocyte allele) are plotted as dark red triangles. The numbers of misregulated genes above and below the diagonal line are indicated. **d** Boxplots showing sperm-specific (sp) and oocyte-specific (oo) log$_2$ fold changes of significantly upregulated genes (34 genes sig up) and significantly downregulated genes (44 sig down). Downregulated genes (blue) are similarly downregulated from sperm and oocyte alleles, while upregulated genes (red) are significantly more upregulated from sperm alleles than oocyte alleles (i.e., pairwise comparisons of transcript reads from sperm vs. oocyte alleles indicate that sperm allele reads are significantly more numerous than oocyte allele reads for upregulated genes but not for downregulated genes). Each box extends from the 25th to the 75th percentile, with the median indicated by the horizontal line; whiskers extend to the most extreme data point which is no more than 1.5 times the interquartile range from the box; the individual data point outside of these boundaries is shown as a filled box. The waist indicates the 95% confidence interval for the medians. *p*-values were generated using the paired student's *t*-test

*M+P−* germlines for the germline proteins PGL-1 (a component of germ granules)[26] and HTP-3 (a component of the synaptonemal complex)[27]. We found that PGL-1 and HTP-3 were below detection (or nearly so) in 100% (*n* = 27) and 28% (*n* = 60) of sterile *K27me3 M+P−* germlines analyzed, respectively, regardless of whether or not they expressed the neuronal GFP reporter (Fig. 3d, e). Although only 28% of sterile *K27me3 M+P−* germlines lacked detectable HTP-3 staining, the remaining 72% of germlines analyzed displayed abnormal HTP-3 patterns (Fig. 3d). Of note, germlines expressing the neuronal UNC-119::GFP reporter consistently displayed more severe HTP-3 phenotypes than germlines that were UNC-119::GFP(−) (Fig. 3d).

Consistent with *K27me3 M+P−* germlines adopting a neuronal identity, we readily observed patches of highly concentrated GFP signal in UNC-119::GFP(+) germlines, indicating that the normally syncytial germ cells had cellularized. In some cases, we observed axo-dendritic structures emanating from these bright GFP(+) cells (Fig. 3b, c). Our findings demonstrate that absence of H3K27me3 from the sperm-inherited genome results in the derepression of many somatic genes, especially neuronal genes, predominantly from sperm-inherited alleles. In a sensitized background, lack of H3K27me3 on the sperm-inherited genome results in germlines that transition away from a germline program and toward a neuronal program.

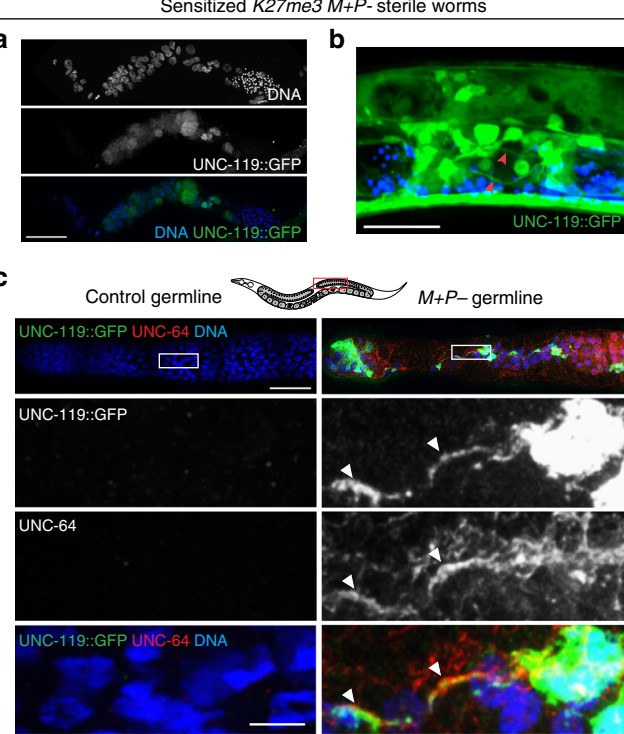

Sensitized *K27me3 M+P-* sterile worms

**a** DNA / UNC-119::GFP / DNA UNC-119::GFP

**b** UNC-119::GFP

**c** Control germline / *M+P−* germline
UNC-119::GFP UNC-64 DNA
UNC-119::GFP
UNC-64
UNC-119::GFP UNC-64 DNA

**Fig. 3** Inheritance of sperm chromosomes lacking H3K27me3 causes germ cells in offspring to transition toward a neuronal fate. **a** Immunofluorescence images of a sterile germline from a *K27me3 M+P−* worm in a sensitized genetic background (see text) and carrying an *unc-119::gfp* neuronal reporter transgene. DNA in blue. GFP in green. Scale bar represents 40 μm. **b** Image of GFP in a live *K27me3 M+P−* worm carrying the *unc-119::gfp* reporter transgene. Arrowheads point to axo-dendritic processes extending from cells in the gonad with high levels of UNC-119::GFP. GFP in green. Blue is autofluorescent gut granules. Scale bar represents 10 μm. **c** Immunofluorescence images of a control germline and a sterile germline from a *K27me3 M+P−* worm showing expression of the UNC-119::GFP reporter in green, immunostaining of the neuronally expressed protein UNC-64 in red, and DNA in blue. Worm schematic indicates the region of germline images. Arrowheads point to axo-dendritic processes co-stained for UNC-119::GFP and UNC-64 in the zoomed-in images. Scale bar in top panel represents 30 μm. Scale bar in lower panel represents 5 μm. **d**, **e** Summary of HTP-3 staining (**d**) and PGL-1 staining (**e**) observed in control germlines from fertile *K27me3 M+Z−; unc-119::gfp* worms and germlines from sterile *K27me3 M+P−* worms. For each staining category, representative immunofluorescence images of DNA and HTP-3 or PGL-1 are shown. Scale bar represents 10 μm (**d**) and 2 μm (**e**)

**d**

Percent of control and sterile *K27me3 M+P-* worms with different patterns of germline-specific protein HTP-3

| | Chromosomal | Reduced chromosomal | Not chromosomal | Nearly below detection |
|---|---|---|---|---|
| Control (n = 31) | 100% | – | – | – |
| M+P− GFP(−/+) (n = 60) | – | 22% | 50% | 28% |
| GFP (−) (n = 31) | – | 42% | 35% | 23% |
| GFP (+) (n = 29) | – | – | 66% | 34% |
| Pachytene germline nuclei | DNA / HTP-3 | | | |

**e**

100% control germlines (n = 12)
DNA / PGL-1 / DNA PGL-1
100% sterile M+P− germlines (n = 27)

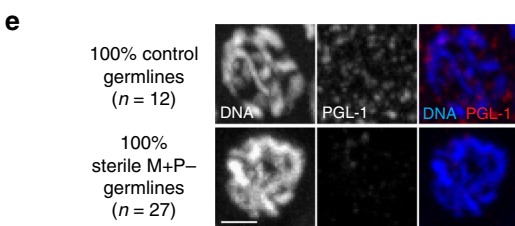

## Discussion

Establishment and maintenance of chromatin states are critical during development and aging. During early development, inherited chromatin states must be rewritten to enable differentiation of different lineages. During aging, established chromatin states must be maintained to preserve cell-type-specific gene expression patterns and thus protect cell identity. Defects in these processes can lead to severe developmental defects and cancer. Whether chromatin states established in the parent and transmitted to offspring via sperm or oocyte influence the establishment and/or maintenance of cell fates in those offspring is a critical question in the field. Our findings demonstrate that in *C. elegans* sperm-inherited chromatin states influence germ cell identity in offspring. Because we observe reprogramming of germ cells toward neuronal cells in adult *K27me3 M+P−* germlines but only rarely in larval *K27me3 M+P−* germlines, we conclude that offspring germ cells lose rather than fail to establish their germ cell identity. These findings reveal that in worms, sperm-inherited H3K27me3 is transmitted to offspring germ cells to prevent germline-inappropriate transcription and thus protect germ cell identity.

Altogether, our findings point to a model in which gamete-inherited patterns of H3K27me3 act as a barrier to inappropriate transcription; in germ cells, this barrier function is particularly important in suppressing transcription of factors that could drive germ cells toward alternative cell identities (Fig. 4). In the germlines of worms that inherited the sperm genome lacking H3K27me3, we speculate that some genes become upregulated as a direct result of diminished H3K27me3 (Factor X in Fig. 4), while other genes may be upregulated as a consequence of being the target of an upregulated gene. An alternative scenario is that Factor X genes are normally expressed in wild-type germ cells but prevented from activating germline-inappropriate target genes by H3K27me3 on those target genes. An example of the first scenario (Fig. 4) is the *vab-7* gene, which encodes a neuronal transcription factor[28]; *vab-7* is not expressed in wild-type germlines but becomes upregulated in *K27me3 M+P−* germlines. Examples of the alternative scenario are genes that encode neuronal specification factors (e.g., *die-1* and *nsy-7*)[28] that are highly expressed in both wild-type and *K27me3 M+P−* germlines. Regardless of the exact cascade of gene activation, our expression analysis of *K27me3 M+P−* germlines suggests a programmatic shift toward a neuronal program and supports the hypothesis that an upstream factor(s) is altering gene expression in a concerted manner, especially from the H3K27me3-depleted sperm genome.

Our findings establish a causal role for sperm-inherited histone marks influencing transcription in offspring and reveal their role in protecting cell identity. Together with observations that in mammals, sperm retain modified histones over developmentally important genes[5–7], this work implicates sperm-inherited chromatin states as one mechanism through which environmentally

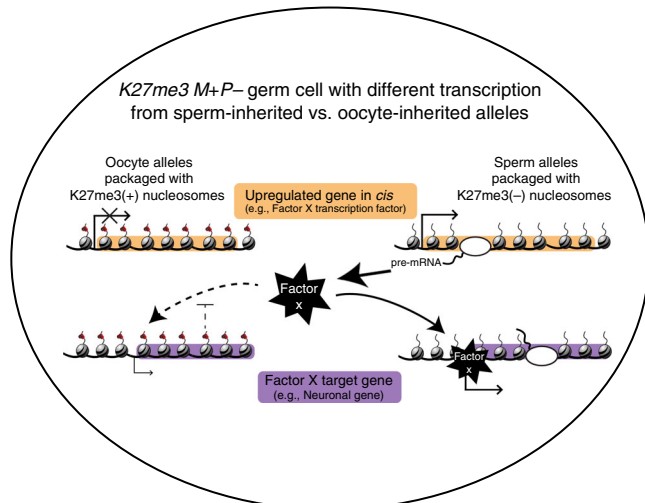

**Fig. 4** A model for transcriptional upregulation in germ cells that inherited sperm chromosomes lacking H3K27me3. The schematic is a germ cell nucleus from a *K27me3 M+P−* worm showing oocyte-inherited and sperm-inherited alleles for 2 representative genes: a transcription factor-encoding gene (orange) and a target gene of that transcription factor (purple). Sperm-inherited alleles lack repressive H3K27me3 and as a result are prone to upregulate Transcription Factor X. If expressed, Transcription Factor X can then activate target genes strongly from H3K27me3(−) sperm alleles and weakly from H3K27me3(+) oocyte alleles. In this model, H3K27me3 would act as a barrier against (1) production of Factor X and (2) activation of Factor X target genes. In *K27me3 M+P+* worms, oocyte-inherited and sperm-inherited alleles are similarly marked with H3K27me3

induced changes to parental epigenomes can impact the development and health of future generations.

## Methods

**C. elegans growth conditions**. Worms were maintained on nematode growth medium (NGM) agar plates spotted with *Escherichia coli* OP50 and maintained at 15 °C (*fem-2* and *plk-1* containing strains) or 20 °C (all other strains). To generate females, *fem-2* strains were raised at 24 °C until L4.

**C. elegans strains**. All strains are in the N2 background except CB4856 and SS1292.

 N2 wild-type isolate from Bristol, England.
 CB4856 wild-type isolate from Hawaii.
 SS1292 *fem-2(b245ts) III* - mutation CRISPR-engineered into CB4856.
 DH0245 *fem-2(b245ts) III*.
 SS1229 *fem-2(b245ts) III; mes-4(bn73) V*.
 SS1167 *mes-3(bn35) I/hT2-GFP (I;III); fem-2(b245ts) III*.
 SS1099 *mes-3(bn35) I/hT2-GFP (I;III); him-8(e1489) IV*.
 OCF65 *plk-1(or683) III; fog-1(e2121) unc-11(e47) I/hT2 [qIs48] (I;III); itIs37 [pAA64: pie-1p::mCHERRY::his-58+unc-119(+)]*.
 SS1424 *mes-3(bn35) I/hT2-GFP(I;III); edIs6[unc-119::GFP+rol-6(su1006)] IV*.
 SS1298 *set-2(bn129) fem-2(b245ts) III*.

**Fixing and staining embryos, larvae, and germlines**. Specimens were flattened between a slide and a coverslip by wicking away buffer (see below). Slides were frozen in liquid nitrogen, coverslips were removed, then slides were fixed in methanol (10 min) and acetone (10 min) and allowed to air dry for up to 3 days. Slides were blocked (1.5% ovalbumin, 1.5% BSA in PBS) for 25 min, then incubated with primary antibodies overnight at 4 °C. Slides were washed 3× in PBST (0.1% tween in phosphate buffered saline pH 7.2), blocked for 5 min, incubated with secondary antibodies and DAPI at room temperature for 2 h, and washed 3× in PBST. Coverslips were mounted on slides with gelutol. Slides were allowed to dry for up to 3 days before being imaged.

**Antibodies**. Primary antibodies and their dilutions were as follows: 1:30,000 of 2.45 mg/mL mouse anti-H3K27me3 (Kimura mAb 1E7 clone CMA323, Wako cat#309-95259), 1:3000 original stock solution of rabbit anti-H3K27me3 (C36B11 Cell Signaling MAb#9733 lot#C36B11), 1:4000 of 0.4 mg/mL mouse anti-GFP (Roche cat#11 814 460 001 lot#14158300), 1:4000 rabbit anti-UNC-64 serum[22],

1:50,000 rabbit anti-PGL-1 serum[23], 1:500 guinea pig anti-HTP-3 serum[24]. Secondary antibodies conjugated to Alexa Fluor: 488 goat anti-mouse (Life cat#A11001), 488 goat anti-rabbit (Molecular Probes cat#A-11008), 594 goat anti-rabbit (Life cat#A11012), and 594 goat anti-guinea pig (Molecular Probes cat#A11076) were used at 1:300 with 0.05 μg/mL 4′,6-diamidino-2-phenylindole (DAPI).

**Microscope set-up for immunocytochemistry**. All images were acquired using the following: Yokogawa CSUX-1 spinning disk scanner, Nikon (Garden City, NY) TE2000-E inverted stand, Hamamatsu ImageEM X2 camera, solid state 405, 488, and 561 nm laser lines, 435–485, 500–550, and 573–613 fluorescent filters, Nikon Plan Apo VC 60×/1.40 oil objective, Nikon Plan Apo 100×/1.40 oil objective, and Micro-Manager software.

**Analysis of MES-4-depleted and SET-2 mutant embryos**. *plk-1; fog-1 unc-11* females (Fig. 1b, c) or *fem-2(ts)* or *set-2 fem-2(ts)* females (Supplementary Fig. 1) were mated to N2 (P+) or *mes-3 M-Z-; him-8* (P−) males (M- for no maternal load of MES-3, Z- for no zygotic synthesis of MES-3) on plates with *mes-4* RNAi, empty vector RNAi, or OP50 (Supplementary Fig. 1c) and incubated overnight at 23 °C (Fig. 1b, c) or 20 °C (Supplementary Fig. 1). Embryos were dissected from mated females in egg buffer (25 mM HEPES pH 7.4, 118 mM NaCl, 48 mM KCl, 2 mM CaCl₂, 2 mM MgCl₂) with 1 mM levamisole on polylysine-coated slides. Embryos attached to slides were fixed and stained as described above. Stacks of optical sections were acquired on a Solamere spinning disk confocal system as described above. Stacks were collapsed into maximum intensity projections using Micro-Manager. Single-channel and merge images were generated in Photoshop. To assess the appearance of H3K27me3 on sperm chromosomes in embryos with and without a maternal load of MES-4, optical sections of *K27me3 M+P-* embryos from *fem-2(ts)* mothers (with a maternal load of MES-4) or from *fem-2(ts); mes-4(bn73)* mothers (lacking a maternal load of MES-4) were stained for DNA and H3K27me3 and scored for: 1) undetectable H3K27me3 on some chromosomes, 2) reduced levels of H3K27me3 on some chromosomes relative to other chromosomes, or 3) indistinguishable levels of H3K27me3 on all chromosomes.

**Analysis of K27me3 M+P− and M+P+ L1 larvae**. *fem-2(ts)* females were mated to N2 (P+) or *mes-3 M-Z-; him-8* (P−) males for ~48 h. Gravid females were transferred to 100 μL S Basal (5.85 g NaCl, 1 g K₂HPO₄, 6 g KH₂PO₄, 1 mL cholesterol (5 mg/mL in ethanol), H₂0 to 1 L) on gelatin chrom alum-coated slides and incubated overnight in a humid chamber at 20 °C. Starved *K27me3 M+P−* and *K27me3 M+P+* L1s were transferred to polylysine-coated slides. L1s attached to slides were fixed and stained as described above. Stacks of optical sections were acquired on a Solamere spinning disk confocal system as described above. Stacks were collapsed into maximum intensity projections using Micro-Manager. Single-channel and merge images were generated in Photoshop.

**Analysis of sensitized K27me3 M+P− and M+P+ germlines**. *mes-3/hT2-GFP; fem-2(ts)* females were mated to either *mes-3/hT2-GFP; unc-119* (P+) or *mes-3 M+Z−; unc-119* (P−) males to generate *K27me3 M+P+* and *K27me3 M+P−* worms, respectively. 146 L4 *K27me3 M+P−* larvae were cut at the pharynx to extrude their anterior germline for assessment of UNC-119::GFP expression in germ cells. Sterile adults on day 1 of adulthood (n = 267) were moved to new plates for live analysis to determine whether germ cells expressed UNC-119::GFP. Sterile day 1 adults that lacked germline expression of UNC-119::GFP were allowed to age at 20 °C for a repeat analysis of UNC-119::GFP expression in germ cells on day 2 of adulthood. L4s, day 1 sterile adults, and day 2 sterile adults were analyzed for germ cell expression of UNC-119::GFP on a Leica MZ16F microscope equipped with a GFP filter. Germlines were imaged for GFP expression both by live imaging of whole worms and by fixing dissected germlines in egg buffer with 1 mM levamisole and 0.5% tween on polylysine-coated slides and immunostaining for GFP, UNC-64, PGL-1, and/or HTP-3. Stacks of optical sections were acquired on a Solamere spinning disk confocal system controlled by Micro-Manager software as described above. Stacks were collapsed into maximum intensity projections using Micro-Manager and stitched together into composite images using Photoshop. Single-channel and merge images were generated in Photoshop. In some cases (Fig. 3c), germlines were straightened using ImageJ.

**Sterility analysis of K27me3 M+P− and M+P+ offspring**. *fem-2(ts)* females were mated to *mes-3/hT2-GFP; fem-2* (P+) or *mes-3 M+Z−; fem-2* (P−) males at 15 °C. *mes-3/+; fem-2 K27me3 M+P+* and *M+P−* L4s (n = 30 replicate 1; n = 40 replicate 2) were cloned to individual plates and allowed to lay eggs for ~24 h, then transferred to fresh plates each day until no more embryos were laid. Embryos from *K27me3 M+P+* and *M+P−* hermaphrodites were incubated at 15 °C until day 1 adulthood, then hermaphrodites were scored as sterile (no embryos in the uterus) or fertile (n = 9370 replicate 1; n = 9693 replicate 2).

**RNA-seq of K27me3 M+P− and M+P+ hybrid and non-hybrid germlines**. Feminized *fem-2* (CB4856 (hybrid) or N2 (non-hybrid) background) hermaphrodites were mated to either *mes-3/hT2-GFP; him-8* (P+) or *mes-3 M-Z-; him-8*

males (P−) to generate genetically identical *mes-3/+*; *fem-2/+*; *him-8/+* *K27me3* *M+P+* and *M+P−* offspring, respectively. In addition, feminized *fem-2* (CB4856 background) were mated to wild-type N2 males to generate "wild-type" hybrid offspring. *mes-3/+*; *fem-2/+*; *him-8/+*(*K27me3* *M+P+* and *M+P−*) and *fem-2/+* ("wild-type" hybrid) L4 offspring were picked and allowed to age overnight to generate *K27me3* *M+P+*, *K27me3* *M+P−*, and "wild-type" hybrid day 1 adults. ~50 germlines were dissected from day 1 adults in egg buffer with 1 mM levamisole, 0.5% tween and transferred into 300 μL of ice-cold TRIzol reagent (Life Technologies), then stored at −80 °C. Samples were subjected to 3 freeze-thaw cycles. One microliter linear polyacrylamide was added to each sample, vortexed, then transferred to Phase Lock Gel-Heavy 2 mL tubes to extract the aqueous phase. 0.75 volume isopropanol was added to each sample and incubated overnight at −20 °C. RNA was isolated via ethanol precipitation and rehydrated in 14.5 μL RNase-free $H_2O$. 2.5 μL were used to assess RNA quality (Agilent RNA Nano Bioanalyzer Chip) and concentration (Qubit Quant-IT RNA Assay Kit). The remaining 12 μL were depleted of rRNA (NEBNext rRNA Depletion Kit) and used for library preparation (NEBNext Ultra RNA Library Prep Kit). Libraries were sequenced on an Illumina HiSeq4000 to acquire paired-end 100 bp reads. Four biological replicates each were generated for *K27me3* *M+P−*, *K27me3* *M+P+*, and "wild-type" hybrid samples.

**Processing and analyzing RNA-seq data.** For differential expression analysis, raw sequences were mapped to transcriptome version WS220 using TopHat2[29]. Only reads with one unique mapping were allowed. Otherwise default options were used. Reads mapping to ribosomal RNAs were removed from further analysis. HTSeq[30] was used to obtain read counts per transcript. DESeq2[31] was used to determine differentially expressed genes from HTSeq counts. A Benjamini–Hochberg multiple hypothesis corrected *p*-value cutoff of 0.1 was used as a significance cutoff to identify significantly differentially expressed genes between *K27me3* *M+P−* and *M+P+* germlines.

To identify RNA-sequencing (RNA-seq) reads coming from the Hawaiian or Bristol parent in the hybrid offspring germlines, the annotated single nucleotide polymorphisms (SNPs) of the Hawaiian strain CB4856 genome compared to the Bristol strain N2 WS220 genome were downloaded from WormBase. 126,615 unique SNPs were downloaded, of which 25,980 SNPs mapped to an exon in a total of 9146 coding genes, according to WormBase. Henceforth, we refer to these SNPs as exonic SNPs.

RNA-seq reads from the Bristol strain, Hawaiian strain, and hybrids were mapped to the Bristol (N2) genome version WS220 using TopHat2, once allowing 1 mismatch in a read and a second time not allowing a mismatch. Reads overlapping exonic SNP positions were identified, and HTSeq was used to count these SNP-containing reads for each transcript. Counts per transcript were normalized by the number of total transcript reads, including non-SNP-overlapping reads, for each of the libraries sequenced. An average count per transcript was obtained by averaging the replicates for each of the Bristol, Hawaiian, and hybrid *K27me3* *M+P−* and *M+P+* replicate experiments. A further quality control step was imposed. For the Bristol and Hawaiian RNA-seq data, we calculated a distance score between the mismatch 0 and mismatch 1 read counts for each transcript. For each transcript i in the Hawaiian RNA-seq data, a score $hz_i$ was calculated: $hz_i = (hmm1_i − hmm0_i)/(hmm1_i + hmm0_i)$ where $hmm1_i$ and $hmm0_i$ denote the normalized read counts for transcript i from mapping with a mismatch 1 and with a mismatch 0 to the Bristol genome, respectively. In the same way, we defined $bz_i$, $bmm1_i$, and $bmm0_i$ for transcripts from mapping our Bristol RNA-seq data. For Hawaiian transcript reads from exonic SNPs, we expect $hz_i$ to be close to 1; all reads should have mapped when allowing 1 mismatch, and none should have mapped when allowing no mismatches. For Bristol transcript reads from exonic SNPs, we expect $bz_i$ to be close to 0, as the number of reads from both mappings should have been very similar. We removed transcripts that did not follow these requirements. If a transcript i had a $hz_i < 0.75$ for the Hawaiian RNA-seq data, or an absolute value $|bz_i| > 0.25$ for the Bristol RNA-seq data, that transcript was removed from further analysis. These requirements removed 1080 transcripts with exonic SNPs. For the remaining 8066 transcripts we obtained from the hybrid *K27me3* *M+P−* and *M+P+* RNA-seq data, we determined the transcript reads from the sperm allele (Bristol) as reads that mapped when allowing 0 mismatches. We determined the transcript reads from the oocyte allele (Hawaii) as reads that mapped when allowing 1 mismatch but that did not map when allowing 0 mismatches. We thus calculated transcript reads from the oocyte allele by subtracting reads that mapped when allowing 0 mismatches (sperm allele reads) from reads that mapped when allowing 1 mismatch (reads from both gamete alleles).

Many genes with exonic SNPs were not expressed in any of our germline RNA-seq data. We filtered out these non-expressed transcripts from further analysis by requiring the normalized read counts from the sperm allele or oocyte allele in at least one condition, *K27me3* *M+P−* or *M+P+*, to be at least 0.5. Because our goal was to use SNP-containing transcript reads as a proxy for total transcript reads, we filtered out transcripts that had a greater than 2.5-fold difference between the $log_2$ of SNP-containing reads versus the $log_2$ of total reads for a given transcript in hybrid germlines (Hawaii mother and Bristol father). This requirement removed 190 genes and left 2,606 transcripts for further analysis. To avoid dividing by zero when calculating log fold changes and to reduce the effect of noise on low read

counts, we added a pseudo-count of 0.5 to all remaining transcripts from oocyte alleles and sperm alleles in both conditions.

**GO analysis of significantly misregulated genes.** Significantly upregulated genes (361 genes) and downregulated genes (342 genes) in the "wild-type" hybrid vs *K27me3* *M+P−* comparison (FDR < 0.05) were used to perform GO analysis (Supplementary Fig. 2b, c). GO analysis was performed using the tissue enrichment tool provided by WormBase (version WS266).

**Statistics.** Sample sizes and statistical tests used are described in the figure legends. Paired student's *t*-tests and Wilcoxon rank sum tests were performed using R. Four biological replicates of sequencing data were used. No replicates were omitted. Differential expression analysis was performed using DESeq2, which uses negative binomial generalized linear models. Adjusted *p*-values were calculated using the Benjamini-Hochberg method performed by DESeq2.

**Reporting Summary.** Further information on experimental design is available in the Nature Research Reporting Summary linked to this article.

## Data availability
Raw sequence files and processed data sets generated in this study are available at the Gene Expression Omnibus (GEO) repository, accession GSE123415. Processed data sets available at GEO are as follows: RNA-seq read counts for *K27me3* *M+P−*, *K27me3* *M+P+* and "wild-type" hybrid containing (1) all read counts mapping to a transcript, (2) read counts overlapping SNPs mapped to the Bristol genome allowing 1 mismatch (Hawaii & Bristol reads), (3) read counts overlapping SNPs mapped to the Bristol genome allowing 0 mismatches (Bristol-specific reads), (4) differential expression analysis comparing transcript reads from *K27me3* *M+P−* vs. *M+P+*, and (5) differential expression analysis comparing transcript reads from *K27me3* *M+P−* vs. "wild-type" hybrid. Supplementary Data 1 consists of sperm-specific and oocyte-specific reads and the $log_2$ fold changes in *K27me3* *M+P−* and *M+P+* for significantly misregulated SNP-containing genes. All other relevant data supporting the key findings of this study are available within the article and its Supplementary Information files or from the corresponding author upon reasonable request. A reporting summary for this Article is available as a Supplementary Information file.

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

## Acknowledgements

We thank Abby Dernburg for HTP-3 antibody, Ben Abrams for imaging advice, and past and present Strome lab members for feedback and helpful discussions. Some strains were provided by the CGC, which is funded by NIH P40 OD010440. This work used the Vincent J. Coates Genomics Sequencing Laboratory at UC Berkeley, supported by NIH S10 OD018174. The research conducted in this study was supported by NIH T32 GM008646, NIH F31 GM120882, and a STARS re-entry scholarship sponsored by Marilyn C. Davis & the American Association of University Women to K.R.K. and by NIH R01 GM34059 to S.S. The content is solely the responsibility of the authors and does not necessarily represent the official views of the National Institutes of Health.

## Author contributions

K.R.K. and S.S. designed the experiments, K.R.K. performed the experiments, A.R. processed RNA sequencing data, K.R.K. and A.R. performed bioinformatic analysis, K.R.K. and S.S. wrote the manuscript.

## Additional information

**Competing interests:** The authors declare no competing interests.

