## [Peer Review File · Nature Communications]

REVIEWERS' COMMENTS:

Reviewer #1 (Remarks to the Author):

Excellent paper, I have very few comments, I recommend to accept it as is. This is another great contribution from the Strome lab that advance our understanding of heritable epigenetic inheritance.

Specifically: Kaneshiro et al., examine the inheritance of the H3K27me marks from the male sperm and the transcriptional consequences of this epigenetic inheritance. The authors use a special and very elegant genetic system (plk-1 mutants) to easily follow sperm-derived chromatin in developing oocyte and provide experimental evidence that show (what was hypothesized before) that MES-4-dependent H3K36me antagonizes de-novo deposition of H3K27me marks on H3K27me(-) chromosomes during embryonic development. Next, the authors utilize an elegant SNP-analysis RNA-seq pipeline to examine the effects in cis and in trans of H3K27me-depleted sperm on the progeny's germline transcription. Using this system, the authors find that specific genes which derive from parental origin tend to be up-regulated in cis, while another group of genes is down-regulated in trans (both maternal and paternal alleles are affected). Lastly, the authors show that the group of the up-regulated genes is enriched with neuronal genes and find that in a sensitized background (reduced maternal H3K27me), the lack of sperm-derived H3K27me leads to loss of germline-cell identity (differentiation into neuronal tissue) and sterility.

Comments:

*1. Regarding Figure 1 - When testing the ability of H3K36me to antagonize H3K27me deposition on sperm-derived H3K27me (-) chromosomes, the authors focus on knock-down of the mes-4 gene. Could the authors test also the effects of the additional H3K36me methyltransferase gene met-1?

In addition, some numbers would help (maybe i missed it): the authors do not supply the number of animals analyzed or any measurement of what percentage of embryos analyzed showed the phenotype.

The legend of colors for figure 1B is missing (one can derive it from 1C, but it is not clear)

**2. Regarding Figure 2. The authors hypothesize that the genes upregulated from the sperm-derived alleles are related to the loss of H3K27me₃ from sperm. This hypothesis is easy to test. The Strome lab recently published an analysis of sperm H3K27me₃, testing the overlap between genes marked by H3K27me₃, and genes upregulated due to sperm (which lacks H3K27me) should be straightforward, and would support the suggested model.

3. Regarding Figure 3, I was wondering, whether the neuronal tested genes unc-119 and unc-64 are amongst the neuronal upregulated genes?

4. Regarding Figure 4, the hypothesized model notes that a stochastically miss-expressed transcription factor X is behind the upregulation of neuronal genes in the germline of the progeny derived from H3K27me(-) sperm. The stochastic miss expression of genes does not necessarily involves miss-expression of a transcription factor and may be the direct result of lack of H3K27me₃ repression. Maybe it's worth discussing?

Oded

Reviewer #2 (Remarks to the Author):

In this study the authors investigate heritable, paternally-derived histone modifications, specifically H3K27me₃, and their role in zygotic chromatin assembly and germline gene expression

in the offspring. They show that the offspring that arise from sperm with chromatin lacking H3K27me3 are unable to add this mark to the chromatin for a number of cell divisions, and that the mark remains off of the chromatin during the establishment of the primordial germ cells. They further show that the prevention of the de novo addition of H3K27me3 in the embryonic chromatin is dependent on mes-4, an H3K36 methyltransferase required to maintain parental germline patterns of H3K36me3 in the offspring. In contrast, the loss of H3K4me3 does not affect the block to de novo H3K27me3 addition. They then investigate the consequences of loss of paternal H3K27me3 and show that germline development is stochastically defective, resulting in an increase in sterility among the offspring derived from sperm lacking H3K27me3. This sterility appears to arise from a loss of germ cell identity, as these offspring show a low but significant penetrance of ectopic expression of a neuronal reporter gene and an endogenous neuron-specific protein, as well as decreased appearance of germ cell specific proteins. The authors thus propose, and I think reasonably conclude, that the role of H3K27me3 in the germline of *C. elegans* is to maintain a chromatin state compatible with germ cell development, rather than to re-establish this state at each generation.

This is an excellent study combining the clever use of genetics and genomic analyses in a set of experiments that address the questions being asked in a straightforward yet elegant manner. I have only a couple of minor comments:

a) In line 97 and in other places they use the phrase “genetically identical” to describe the strains that are crossed for SNP analyses in the offspring. This seems too strong a term for strains that, by definition, differ in genome sequence. Perhaps initially defining the strains using some other phrase, such as “genotypes differing only in a number of SNPs” and then using the strain names as the SNP identifiers?

b) I am curious what the authors think about why neuronal gene expression seems to be the easiest default ectopic expression in this and the studies of others showing ectopic somatic expression in germ cells. For example, the *pie-1* mutation causes premature transcriptional activation in germ cell precursors, but they differentiate into muscle and intestine, not neurons. Why wouldn't the PRC2 mode of repression prevent this?

c) Can the authors perhaps also comment on why PRC2 and MES-4 are required for the ectopic expression of somatic genes in a number of repressor mutants—how does this study inform those?

Reviewer #3 (Remarks to the Author):

The paper by Kaneshiro et al describe the influence of paternal epigenetic information, in the form of histone post-translational modification, on development and function of the next generation. The topic is highly interesting but under-studied due to the difficulty of studying this phenomena in many organisms. The authors capitalize on the simple developmental plan of *C. elegans*, as well as the lack of histone methylation, to focus on the role of H3K27me3 that is passed on paternally to new offspring. They employ a combination of genetics, cytology, and gene expression studies to show that chromosome paternally marked with H3K27me3 function in regulating the expression of genes important for germline fate or function in offspring. This finding is novel. The experiments are exceptionally well done as well. As such, the topic of the paper and main findings would be of great interest to the readers of Nature Communications. The public at large would also be intrigued by evidence that epigenetic contributions from each parent have direct influence on the fertility of offspring.

I had some questions regarding the manuscript as presented that could be addressed by the authors.

1) The first is: How far in development of the germline can the authors still detect differences in H3K27me3 in the M+P- animals? This is because they conduct transcriptomic analysis on adult germlines and find clear differences – when do those differences arise and can they be attributed to differences in H3K27me3 status in the adult germline? The authors have evidence of this in a previous paper from Gaydos 2014. There is a gap in the present paper that doesn't connect the staining pattern seen in the PGCs of larvae and expression differences in adult germlines that should be better explained.

2) There are some questions about the gene expression differences the authors find. First, of the genes identified that show differences in expression – how consistent are these changes across replicates? The authors use the word “stochastic” to describe the differences and it is unclear what that means. Second – I find figure 2D difficult to interpret (also Supplemental Figure 2A).

3) Overall, there is a lack of clear explanation or wording in places that are listed in the detailed comments below. This makes the interpretation of some of the data – particularly Figure 2D and Figure 3 (with the 'sensitized background) difficult. I expect this may be due to trying to adhere to word limits. Please expand on the explanation to make the rationale for experiments and data interpretation more clear to non-specialists that would be reading Nature Communications.

Detailed comments:

Abstract;

Line 8: The authors allude to “growing medical relevance” but do not explain or make concrete what that is later in the manuscript.

Line 12: This sentence is hard to understand (the use of 'antagonism' is not clear): “We show that sperm chromosomes inherited lacking the repressive histone modification H3K27me3 are maintained in that state by H3K36me3 antagonism.”

Line 19: change “through which epigenetic information can be passed from father to shape” to “through which epigenetic information from a father can shape..”

Line 80: The following sentence is very long. It is difficult to follow.

“This gradual acquisition of H3K27me3 matches the gradual loss of H3K36me3 in the absence of maternal MES-4 and demonstrates that maintenance of the H3K27me3(-) state during embryogenesis requires that PRC2 activity be antagonized by MES-4, likely through the histone mark it generates.”

A suggestion is to make the relationships more direct: “This gradual acquisition of H3K27me3 matches the gradual loss of H3K36me3 in the absence of maternal MES-4. Thus, MES-4 antagonizes [or inhibits?] PRC2 activity, presumably through the H3K36me3 mark that MES-4 generates, to maintain the paternal H3K27me3(-) state.”

Line 103-108: The authors show PGC of larvae in Figure 1D and then quickly transition to discuss conducting transcriptional profiling to determine “differences in inherited chromatin states ... in the developing germline.” What stage of 'development' are the germlines that they are analyzing? Is there anything known about the gamete-inherited states as PGCs divide to form the germline in larvae (cytologically?). There is a leap here from showing the PGCs to then addressing the consequences of fully formed germlines in adults. One question is whether or not there are differences in histone modification states in adult germlines. If not, how do differences in transcriptional state then arise?

Line 110 “149 genes were upregulated and 116 genes were downregulated”

Of these genes, how many were identified in more than one replicate? In other words, how consistent was the set of genes that were found as up and down regulated? The authors could show this in a supplemental table that lists all of the genes that showed differences in expression and the replicates they were identified in.

Line 115: In general, the wording of this paragraph is confusing.

For example: "Using SNPs, we could determine for 23% of misexpressed genes whether they were misexpressed from the sperm allele, the oocyte allele, or both."

Does this mean that "Of the misexpressed genes identified, we could use SNPs to determine if expression differences arose from the sperm allele, oocyte allele, or both."?

Line 116: What does this sentence mean "Genes' changes in expression based on SNP-containing reads correlate well with changes based on all reads mapping to the genes ($r=0.6$, $p<2.2e-16$)"?

Line 120: It is difficult to easily see from Figure 2C and D how the authors reach this interpretation "Our analysis of SNP-containing reads indicated that downregulated genes reflect decreased transcription from both the sperm and oocyte allele. However, upregulated genes reflect increased transcription primarily from the sperm allele (Fig. 2C, 2D). Importantly, half of these genes are upregulated in cis with increased transcription from the sperm allele specifically (at least 1.5 fold increased transcription from the sperm allele but not the oocyte allele)."

-Can the authors make more direct links to features of the figures? They list numbers of genes in these categories in the figure – can they say which corresponds to which? For example "Our analysis of SNP-containing reads indicated that downregulated genes reflect decreased transcription from both the sperm and oocyte allele (44 genes total). Importantly, half of these genes (half of what? How many? How does that correspond to the numbers in the figure?) are upregulated in cis with increased transcription from the sperm allele (how many?) specifically (at least 1.5 fold increased transcription from the sperm allele but not the oocyte allele)
It is difficult to interpret Figure 2D– what are log fold changes indicating?

Line 126: "this finding establishes a causal role for sperm-inherited H3K27me3 in regulating transcription in offspring."

-'establishes a causal role' is strong. "Suggests" is suitable.

Line 128: This line is speculative at this point in the paper: "It also raises the possibility that environmentally induced changes."

Line 130: "The observation that K27me3 M+P- worms have altered transcription from oocyte alleles is not surprising and likely represents secondary effects of increased transcription from sperm alleles."

This sentence is unnecessary. The next sentence is more clear and better explained.

Line 145: "carrying a pan-neuronal reporter gene, *unc-119::GFP*."

Is this at the endogenous locus?

Also, was this one of the genes that showed altered expression? Was *UNC-64*?

More explanation of the list of genes would help to reveal whether the effects they see are direct or indirect.

Line 148-151: The info in this paragraph becomes confusing. The authors first saw both M+P+ and M+P- worms are fertile, but then talk about a "sensitized background in which the maternal load of PRC2 is reduced by half" means. This is quite perplexing. Do they mean the original *mes-3* mutation that they use to create the M+P-? How is this further sensitized? In general, how the experiment was conducted and the interpretation of the data (below) thus becomes quite confusing.

Line 152-156: The authors say that 870 worms were examined but then indicate numbers below that say $n=894$. It is unclear how the numbers of worms discussed in this paragraph correspond to one another. It is also not clear from the figure or figure legend for Figure 3 (there are several panels but the authors call out "Fig.3" in the text).

Line 189: "Because we observe reprogramming of germ cells toward neuronal cells in adult

K27me3 M+P- germlines but only rarely in larval K27me3 M+P- germlines, we conclude that offspring germ cells lose rather than fail to establish their germ cell identity."

-It is unclear what the authors mean by 'only rarely in larval K27me3 M+P- germlines'. Did they show analysis of larval germlines for neuronal markers?

Line 199: "We speculate that these factors become stochastically expressed in worms that inherit sperm chromatin lacking H3K27me3."

-What do the authors mean by 'stochastically' here? Are the factors random? Or how the factors they identified randomly expressed? Or do they mean variable?

Figure 1A: For the F1 embryo, the picture shows two nuclei – is this depicting a one cell embryo?

Figure 1B: Please include what is labeled in green and red. For the 'single nucleus per cell' is that in a two cell embryo (since there are two nuclei)? For the 'pair of nuclei per cell' is that also a two cell embryo? It is difficult to interpret from the figure, figure legend or text how to interpret the lower left hand image since this is P- and thus something should not be stained? Also, what is the genotypes of the embryos? Are they all the plk mutant?

Figure 2 B – what does the small circle above each condition represent?

Figure 2D – This graph is difficult to interpret. What does log₂ fold change indicate? It is also unclear what the sp and oo mean in terms of the sig up and sig down classes.

Figure 3A and B – What do M+P+ worms look like in comparison to these panels? Do non-sterile worms also show UNC-119 signal?

Figure 3D: What is being quantified in the table? Please label the table.

Figure 4: It is difficult to follow this figure because it is only showing the M+P- scenario. Not sure what the 'derepressed gene in cis' means. The cis and trans aspects of this figure are not well described in the text and thus are confusing when looking at this figure.

Supplemental Figure 2 – It is difficult to follow the hybrid and non-hybrid explanations from the methods and this figure to determine exactly how to interpret this figure. In general, the authors don't really explain hybrid and non-hybrid explicitly. Perhaps a diagram of the crosses would be helpful to explain this.

Reviewer #1 (Remarks to the Author):

Excellent paper, I have very few comments, I recommend to accept it as is. This is another great contribution from the Strome lab that advance our understanding of heritable epigenetic inheritance. Specifically: Kaneshiro et al., examine the inheritance of the H3K27me marks from the male sperm and the transcriptional consequences of this epigenetic inheritance. The authors use a special and very elegant genetic system (plk-1 mutants) to easily follow sperm-derived chromatin in developing oocyte and provide experimental evidence that show (what was hypothesized before) that MES-4-dependent H3K36me antagonizes de-novo deposition of H3K27me marks on H3K27me(-) chromosomes during embryonic development. Next, the authors utilize an elegant SNP-analysis RNA-seq pipeline to examine the effects in cis and in trans of H3K27me-depleted sperm on the progeny's germline transcription. Using this system, the authors find that specific genes which derive from parental origin tend to be up-regulated in cis, while another group of genes is down-regulated in trans (both maternal and paternal alleles are affected). Lastly, the authors show that the group of the up-regulated genes is enriched with neuronal genes and find that in a sensitized background (reduced maternal H3K27me), the lack of sperm-derived H3K27me leads to loss of germline-cell identity (differentiation into neuronal tissue) and sterility.

Comments:

*1. Regarding Figure 1 - When testing the ability of H3K36me to antagonize H3K27me deposition on sperm-derived H3K27me (-) chromosomes, the authors focus on knock-down of the mes-4 gene. Could the authors test also the effects of the additional H3K36me methyltransferase gene met-1?

We chose to focus on the H3K36me3 methylator MES-4 because it is responsible for H3K36me3 maintenance during early embryogenesis; absence of MES-4 during early embryogenesis results in progressive disappearance of H3K36me3 by immunostaining, while absence of MET-1 does not impact H3K36me3 immunostaining in early embryos (Kreher et al. 2018; ref 23). For this reason, loss of maternal MET-1 is not predicted to antagonize PRC2 activity via maintenance of H3K36me3 on H3K27me3(-) chromosomes.

In addition, some numbers would help (maybe i missed it): the authors do not supply the number of animals analyzed or any measurement of what percentage of embryos analyzed showed the phenotype.

We added quantification of the H3K27me3 status of sperm chromosomes in early embryos with and without maternal MES-4. See lines 92-99.

The legend of colors for figure 1B is missing (one can derive it from 1C, but it is not clear)

We explained the colors in each panel in the Fig. 1 legend.

**2. Regarding Figure 2. The authors hypothesize that the genes upregulated from the sperm-derived alleles are related to the loss of H3K27me3 from sperm. This hypothesis is easy to test. The Strome lab recently published an analysis of sperm H3K27me3, testing the overlap between genes marked by H3K27me3, and genes upregulated due to sperm (which lacks H3K27me) should be straightforward, and would support the suggested model.

96% of upregulated genes are enriched for H3K27me3 in wild-type sperm (Tabuchi et al. 2018; ref 11). We added that to the main text and added a new supplementary figure highlighting upregulated genes on ChIP-seq plots from wild-type sperm (Tabuchi et al. 2018; ref 11). See line 139 & Supplementary Figure 4.

3. Regarding Figure 3, I was wondering, whether the neuronal tested genes *unc-119* and *unc-64* are amongst the neuronal upregulated genes?

The neuronal genes *unc-119* and *unc-64* are not upregulated in *K27me3 M+P-* fertile germlines. We added that to the text. See lines 181-182.

4. Regarding Figure 4, the hypothesized model notes that a stochastically miss-expressed transcription factor X is behind the upregulation of neuronal genes in the germline of the progeny derived from H3K27me(-) sperm. The stochastic miss expression of genes does not necessarily involve miss-expression of a transcription factor and may be the direct result of lack of H3K27me3 repression. Maybe it's worth discussing?

It is likely that some genes become derepressed as a direct result of diminished H3K27me3. The patterns of misexpression suggest a programmatic change from a germline program toward a neuronal program. This leads us to speculate that one or more factors are involved in driving this concerted deviation in germline gene expression. See lines 233-246.

Reviewer #2 (Remarks to the Author):

In this study the authors investigate heritable, paternally-derived histone modifications, specifically H3K27me3, and their role in zygotic chromatin assembly and germline gene expression in the offspring. They show that the offspring that arise from sperm with chromatin lacking H3K27me3 are unable to add this mark to the chromatin for a number of cell divisions, and that the mark remains off of the chromatin during the establishment of the primordial germ cells. They further show that the prevention of the de novo addition of H3K27me3 in the embryonic chromatin is dependent on *mes-4*, an H3K36 methyltransferase required to maintain parental germline patterns of H3K36me3 in the offspring. In contrast, the loss of H3K4me3 does not affect the block to de novo H3K27me3 addition. They then investigate the consequences of loss of paternal H3K27me3 and show that germline development is stochastically defective, resulting in an increase in sterility among the offspring derived from sperm lacking H3K27me3. This sterility appears to arise from a loss of germ cell identity, as these offspring show a low but significant penetrance of ectopic expression of a neuronal reporter gene and an endogenous neuron-specific protein, as well as decreased appearance of germ cell specific proteins. The authors thus propose, and I think reasonably conclude, that the role of H3K27me3 in the germline of *C. elegans* is to maintain a chromatin state compatible with germ cell development, rather than to re-establish this state at each generation. This is an excellent study combining the clever use of genetics and genomic analyses in a set of experiments that address the questions being asked in a straightforward yet elegant manner. I have only a couple of minor comments:

a) In line 97 and in other places they use the phrase "genetically identical" to describe the strains that are crossed for SNP analyses in the offspring. This seems too strong a term for strains that, by definition, differ in genome sequence. Perhaps initially defining the strains using some other phrase, such as "genotypes differing only in a number of SNPs" and then using the strain names as the SNP identifiers?

These worms are genetically identical. Both *K27me3 M+P+* and *H3K27me3 M+P-* were generated by mating Hawaii mothers (*fem-2*) to Bristol fathers (*mes-3* or *mes-3/+*). Their genotype is *hybrid fem-2/+; mes-3/+*, and each contains SNPs associated with Hawaii and Bristol genomes. We added the genotype (*hybrid fem-2/+; mes-3/+*) to the text where we refer to these worms being genetically

identical, and we added a description of “hybrid” to the Fig. 1A legend. See line 118.

b) I am curious what the authors think about why neuronal gene expression seems to be the easiest default ectopic expression in this and the studies of others showing ectopic somatic expression in germ cells. For example, the *pie-1* mutation causes premature transcriptional activation in germ cell precursors, but they differentiate into muscle and intestine, not neurons. Why wouldn't the PRC2 mode of repression prevent this?

A number of observations support a model in which among somatic programs, neural appears to be the default or ground state. This is touched on in (Hobert 2013; ref 28). Some additional references proposing a neural program as a default program are below. See references 1 & 2 below.

c) Can the authors perhaps also comment on why PRC2 and MES-4 are required for the ectopic expression of somatic genes in a number of repressor mutants-how does this study inform those?

We think the reviewer means “why PRC2 and MES-4 are required for the ectopic expression of germline genes in somatic cells in a number of repressor mutants”. We answer that inquiry here but think it does not belong in our paper. ChIP analysis of early embryos indicates that the chromatin landscape reflects parental germline gene expression patterns (Rechtsteiner et al. 2010; ref 13, Gaydos et al. 2012; ref 14). The repressive mark H3K27me3, generated by PRC2, is enriched on genes that were silent in the parental germline, and H3K36me3, generated by MET-1 and MES-4, is enriched on genes that were expressed in the parental germline. PRC2 and MES-4 (not MET-1) are required to maintain levels of inherited H3K27me3 and H3K36me3, respectively, during early embryogenesis. These observations suggest that PRC2 and MES-4 are maintaining inherited chromatin states consistent with germline gene expression patterns in all cells of the early embryo. In worms that lack the repressive DRM complex and other synMuv B proteins, somatic cells ectopically express germline genes in a manner dependent upon PRC2 and MES-4. We envision that this ectopic expression is dependent on early somatic blastomeres inheriting a germline chromatin landscape and thus requires a maternal load of PRC2 and MES-4.

Reviewer #3 (Remarks to the Author):

The paper by Kaneshiro et al describe the influence of paternal epigenetic information, in the form of histone post-translational modification, on development and function of the next generation. The topic is highly interesting but under-studied due to the difficulty of studying this phenomena in many organisms. The authors capitalize on the simple developmental plan of *C. elegans*, as well as the lack of histone methylation, to focus on the role of H3K27me3 that is passed on paternally to new offspring. They employ a combination of genetics, cytology, and gene expression studies to show that chromosome paternally marked with H3K27me3 function in regulating the expression of genes important for germline fate or function in offspring. This finding is novel. The experiments are exceptionally well done as well. As such, the topic of the paper and main findings would be of great interest to the readers of Nature Communications. The public at large would also be intrigued by evidence that epigenetic contributions from each parent have direct influence on the fertility of offspring.

I had some questions regarding the manuscript as presented that could be addressed by the authors.

1) The first is: How far in development of the germline can the authors still detect differences in H3K27me3 in the M+P- animals? This is because they conduct transcriptomic analysis on adult germlines and find clear differences - when do those differences arise and can they be attributed to differences in H3K27me3 status in the adult germline? The authors have evidence of this in a

previous paper from Gaydos 2014. There is a gap in the present paper that doesn't connect the staining pattern seen in the PGCs of larvae and expression differences in adult germlines that should be better explained.

In *K27me3 M+P-* animals, sperm chromosomes start acquiring H3K27me3 soon after the PGCs begin to proliferate (Gaydos et al. 2014; ref 12). During larval development, global levels of H3K27me3 on sperm- and egg-inherited chromosomes become more and more similar based on immunostaining. We do not know about levels of H3K27me3 on individual genes. We speculate that lack of H3K27me3 on sperm alleles during the onset of germ cell transcription and proliferation could result in their being prone to activate expression of inappropriate genes. Once activated, a euchromatic state could, in theory, be propagated at specific loci, sensitizing daughter cells to continued expression of inappropriate genes and potentially their target genes. See lines 122-130.

2) There are some questions about the gene expression differences the authors find. First, of the genes identified that show differences in expression - how consistent are these changes across replicates?

The significantly misregulated genes were identified by a statistical test (Love et al. 2014; ref 31) requiring a False Discovery Rate < 0.1 when all experimental replicates were compared to all control replicates. This, by definition, requires differences that are relatively consistent between experimental replicates and control replicates; otherwise they could not be detected as statistically significant.

The authors use the word "stochastic" to describe the differences and it is unclear what that means. Second - I find figure 2D difficult to interpret (also Supplemental Figure 2A).

All replicates are highly correlated in terms of misexpressed genes. However, each replicate includes the combined expression profiles of ~50 germlines. Our population analyses would fail to detect worm-to-worm or germ cell-to-germ cell differences. The stochastic nature of misexpression is inferred from sterility analysis of F2 worms and sterility analysis and misexpression of neuronal transgenes in sensitized *K27me3 M+P-* F1 worms. The sterility of F2 worms and sensitized F1 worms is stochastic in that not all worms within a brood are sterile. The misexpression of neuronal transgenes is stochastic in that not all sterile *K27me3 M+P-* worms display misexpression of neuronal transgenes. Furthermore, within a germline that misexpresses neuronal markers, some germ cells show misexpression while others do not.

3) Overall, there is a lack of clear explanation or wording in places that are listed in the detailed comments below. This makes the interpretation of some of the data - particularly Figure 2D and Figure 3 (with the 'sensitized background') difficult. I expect this may be due to trying to adhere to word limits. Please expand on the explanation to make the rationale for experiments and data interpretation more clear to non-specialists that would be reading Nature Communications.

Detailed comments:

Abstract;

Line 8: The authors allude to "growing medical relevance" but do not explain or make concrete what that is later in the manuscript.

We added citations to support this statement. See references 1 and 2 below.

Line 12: This sentence is hard to understand (the use of 'antagonism' is not clear): "We show that

sperm chromosomes inherited lacking the repressive histone modification H3K27me3 are maintained in that state by H3K36me3 antagonism."

The use of antagonism in the abstract and in the title for Fig. 1 seems like the best word choice. We switched antagonizes to inhibits in one location in the main text. See lines 15, 101, and Fig.1 legend.

Line 19: change "through which epigenetic information can be passed from father to shape" to "through which epigenetic information from a father can shape.."

We adopted the suggested (better) wording. See line 19.

Line 80: The following sentence is very long. It is difficult to follow.

"This gradual acquisition of H3K27me3 matches the gradual loss of H3K36me3 in the absence of maternal MES-4 and demonstrates that maintenance of the H3K27me3(-) state during embryogenesis requires that PRC2 activity be antagonized by MES-4, likely through the histone mark it generates."

A suggestion is to make the relationships more direct: "This gradual acquisition of H3K27me3 matches the gradual loss of H3K36me3 in the absence of maternal MES-4. Thus, MES-4 antagonizes [or inhibits?] PRC2 activity, presumably through the H3K36me3 mark that MES-4 generates, to maintain the paternal H3K27me3(-) state."

We adopted the suggested (better) wording. See lines 99-100.

Line 103-108: The authors show PGC of larvae in Figure 1D and then quickly transition to discuss conducting transcriptional profiling to determine "differences in inherited chromatin states ... in the developing germline." What stage of 'development' are the germlines that they are analyzing? Is there anything known about the gamete-inherited states as PGCs divide to form the germline in larvae (cytologically?). There is a leap here from showing the PGCs to then addressing the consequences of fully formed germlines in adults. One question is whether or not there are differences in histone modification states in adult germlines. If not, how do differences in transcriptional state then arise?

See our response above and new text on lines 122-130.

Line 110 "149 genes were upregulated and 116 genes were downregulated"

Of these genes, how many were identified in more than one replicate? In other words, how consistent was the set of genes that were found as up and down regulated? The authors could show this in a supplemental table that lists all of the genes that showed differences in expression and the replicates they were identified in.

All were found in more than one replicate. Please see discussion above regarding use of the term "stochastic".

Line 115: In general, the wording of this paragraph is confusing.

For example: "Using SNPs, we could determine for 23% of misexpressed genes whether they were misexpressed from the sperm allele, the oocyte allele, or both."

Does this mean that "Of the misexpressed genes identified, we could use SNPs to determine if expression differences arose from the sperm allele, oocyte allele, or both."?

We clarified the wording. See lines 144-148.

Line 116: What does this sentence mean "Genes' changes in expression based on SNP-containing reads correlate well with changes based on all reads mapping to the genes ($r=0.6$, $p<2.2e-16$)"?

We clarified the wording. See line 144.

Line 120: It is difficult to easily see from Figure 2C and D how the authors reach this interpretation "Our analysis of SNP-containing reads indicated that downregulated genes reflect decreased transcription from both the sperm and oocyte allele. However, upregulated genes reflect increased transcription primarily from the sperm allele (Fig. 2C, 2D). Importantly, half of these genes are upregulated in cis with increased transcription from the sperm allele specifically (at least 1.5 fold increased transcription from the sperm allele but not the oocyte allele)."

-Can the authors make more direct links to features of the figures? They list numbers of genes in these categories in the figure - can they say which corresponds to which? For example "Our analysis of SNP-containing reads indicated that downregulated genes reflect decreased transcription from both the sperm and oocyte allele (44 genes total). Importantly, half of these genes (half of what? How many? How does that correspond to the numbers in the figure?) are upregulated in cis with increased transcription from the sperm allele (how many?) specifically (at least 1.5 fold increased transcription from the sperm allele but not the oocyte allele)

It is difficult to interpret Figure 2D- what are log fold changes indicating?

We changed the color and point label (dark red triangles Fig. 2C) for genes upregulated in *cis*. We also added descriptive text to the Fig. 2 legend.

Line 126: "this finding establishes a causal role for sperm-inherited H3K27me3 in regulating transcription in offspring."

-'establishes a causal role' is strong. "Suggests" is suitable.

The observation that many upregulated genes are upregulated in *cis*, discounting small RNAs as a mediator, and the known absence of canonical DNA methylation in *C. elegans*, together strongly support sperm-inherited H3K27me3 as the cause of the observed gene expression changes. We think this justifies the use of strong wording.

Line 128: This line is speculative at this point in the paper: "It also raises the possibility that environmentally induced changes."

The phrasing "raising the possibility" is meant to indicate speculation.

Line 130: "The observation that K27me3 M+P- worms have altered transcription from oocyte alleles is not surprising and likely represents secondary effects of increased transcription from sperm alleles." This sentence is unnecessary. The next sentence is more clear and better explained.

We agree and removed the sentence in question. See line 159.

Line 145: "carrying a pan-neuronal reporter gene, *unc-119::GFP*."
Is this at the endogenous locus?

This reporter is not at the endogenous locus. It is referred to as a transgene in the paper. See Fig. 3 legend.

Also, was this one of the genes that showed altered expression? Was UNC-64?
More explanation of the list of genes would help to reveal whether the effects they see are direct or indirect.

Neither transcript is significantly upregulated in the *K27me3 M+P-* fertile germline data presented in this paper. Transcript analysis of sensitized *K27me3 M+P-* germlines (not carrying the *unc-119::gfp* transgene) (Tabuchi et al. 2018; ref 11) indicates that both trend up, but do not achieve our cut-off for significance.

Line 148-151: The info in this paragraph becomes confusing. The authors first say both M+P+ and M+P- worms are fertile, but then talk about a "sensitized background in which the maternal load of PRC2 is reduced by half" means. This is quite perplexing. Do they mean the original *mes-3* mutation that they use to create the M+P-? How is this further sensitized? In general, how the experiment was conducted and the interpretation of the data (below) thus becomes quite confusing.

We included the genotype of the mother in parentheses to clarify that the reduced maternal load of PRC2 is due to the mother being heterozygous for *mes-3*, which encodes a required component of PRC2. In a non-sensitized background, the mother is homozygous wild type for all PRC2 components. See line 183.

Line 152-156: The authors say that 870 worms were examined but then indicate numbers below that say n=894. It is unclear how the numbers of worms discussed in this paragraph correspond to one another. It is also not clear from the figure or figure legend for Figure 3 (there are several panels but the authors call out "Fig.3" in the text).

We simplified the text and instead added this information as a supplementary table to clarify genotypes, numbers, and percentages of worms analyzed and expressing *unc-119::gfp*. See Supplementary Table 1 and lines 184-189.

Line 189: "Because we observe reprogramming of germ cells toward neuronal cells in adult *K27me3 M+P-* germlines but only rarely in larval *K27me3 M+P-* germlines, we conclude that offspring germ cells lose rather than fail to establish their germ cell identity."
-It is unclear what the authors mean by 'only rarely in larval *K27me3 M+P-* germlines'. Did they show analysis of larval germlines for neuronal markers?

We simplified the text and instead added this information as a supplementary table to clarify genotypes, numbers, and percentages of worms analyzed and expressing *unc-119::gfp*. See Supplementary Table 1 and lines 187-193.

Line 199: "We speculate that these factors become stochastically expressed in worms that inherit sperm chromatin lacking H3K27me3."
-What do the authors mean by 'stochastically' here? Are the factors random? Or how the factors they identified randomly expressed? Or do they mean variable?

We removed "stochastic" from this text and instead included discussion to clarify what we mean. We also discuss above our reasoning for using "stochastic" (see response to initial comment #2 above).

Figure 1A: For the F1 embryo, the picture shows two nuclei - is this depicting a one cell embryo?

That is a picture of a 1-cell embryo showing the maternal and paternal pronuclei. To clarify, we added that information to the Fig. 1 legend.

Figure 1B: Please include what is labeled in green and red. For the 'single nucleus per cell' is that in a two cell embryo (since there are two nuclei)? For the 'pair of nuclei per cell' is that also a two cell embryo? It is difficult to interpret from the figure, figure legend or text how to interpret the lower left hand image since this is P- and thus something should not be stained? Also, what is the genotypes of the embryos? Are they all the plk mutant?

All the embryos in panel b are 2-cell. We added a title to this panel indicating that they are 2-cell embryos. We also included a description in the Fig. 1 legend.

Figure 2 B - what does the small circle above each condition represent?

Small circles represent outliers. We added that description to the Fig. 2B legend.

Figure 2D - This graph is difficult to interpret. What does log2 fold change indicate? It is also unclear what the sp and oo mean in terms of the sig up and sig down classes.

We added more description to the Fig. 2D legend to clarify.

Figure 3A and B - What do M+P+ worms look like in comparison to these panels? Do non-sterile worms also show UNC-119 signal?

We found that fertile worms do not express the *unc-119::gfp* transgene in their germlines. Even in a sensitized background, *K27me3 M+P+* worms are very rarely sterile (<0.3%). We did not observe *unc-119::gfp* expression in the *K27me3 M+P+* worms analyzed. See Supplementary Table 1 and lines 180-190.

Figure 3D: What is being quantified in the table? Please label the table.

The table shows percentage of different HTP-3 phenotypes in control and *K27me3 M+P-* germlines. We added a title to the table in Fig. 3D to convey this.

Figure 4: It is difficult to follow this figure because it is only showing the M+P- scenario. Not sure what the 'derepressed gene in cis' means. The cis and trans aspects of this figure are not well described in the text and thus are confusing when looking at this figure.

We changed 'derepressed' to 'upregulated' in Fig. 4. We also added a description in the Fig. 4 legend of how this model plays out in *K27me3 M+P+* (or wild-type) germ cells.

Supplemental Figure 2 - It is difficult to follow the hybrid and non-hybrid explanations from the methods and this figure to determine exactly how to interpret this figure. In general, the authors don't really explain hybrid and non-hybrid explicitly. Perhaps a diagram of the crosses would be helpful to explain this.

We added a description distinguishing hybrid versus non-hybrid to the Fig.1 legend.

References

1. Hemmati-brivanlou, A. & Melton, D. Will become nerve cells unless told otherwise. **88**, 13–17 (1997).
2. Kamiya, D. *et al.* Intrinsic transition of embryonic stem-cell differentiation into neural progenitors. *Nature* **470**, 503–509 (2011).